# Optimizing Vancomycin Therapy in Critically Ill Children: A Population Pharmacokinetics Study to Inform Vancomycin Area under the Curve Estimation Using Novel Biomarkers

**DOI:** 10.3390/pharmaceutics15051336

**Published:** 2023-04-25

**Authors:** Kevin J. Downes, Athena F. Zuppa, Anna Sharova, Michael N. Neely

**Affiliations:** 1The Center for Clinical Pharmacology, Children’s Hospital of Philadelphia, Philadelphia, PA 19104, USA; 2Clinical Futures, Children’s Hospital of Philadelphia, Philadelphia, PA 19104, USA; 3Division of Infectious Diseases, Children’s Hospital of Philadelphia, Philadelphia, PA 19104, USA; 4Department of Pediatrics, Perelman School of Medicine, University of Pennsylvania, Philadelphia, PA 19104, USA; 5Children’s Hospital Los Angeles, Los Angeles, CA 90027, USA; 6Keck School of Medicine, University of Southern California, Los Angeles, CA 90033, USA

**Keywords:** critical illness, sepsis, kidney injury, biomarkers, pediatric pharmacology, population pharmacokinetics, Bayesian estimation

## Abstract

Area under the curve (AUC)-directed vancomycin therapy is recommended, but Bayesian AUC estimation in critically ill children is difficult due to inadequate methods for estimating kidney function. We prospectively enrolled 50 critically ill children receiving IV vancomycin for suspected infection and divided them into model training (n = 30) and testing (n = 20) groups. We performed nonparametric population PK modeling in the training group using Pmetrics, evaluating novel urinary and plasma kidney biomarkers as covariates on vancomycin clearance. In this group, a two-compartment model best described the data. During covariate testing, cystatin C-based estimated glomerular filtration rate (eGFR) and urinary neutrophil gelatinase-associated lipocalin (NGAL; full model) improved model likelihood when included as covariates on clearance. We then used multiple-model optimization to define the optimal sampling times to estimate AUC_24_ for each subject in the model testing group and compared the Bayesian posterior AUC_24_ to AUC_24_ calculated using noncompartmental analysis from all measured concentrations for each subject. Our full model provided accurate and precise estimates of vancomycin AUC (bias 2.3%, imprecision 6.2%). However, AUC prediction was similar when using reduced models with only cystatin C-based eGFR (bias 1.8%, imprecision 7.0%) or creatinine-based eGFR (bias −2.4%, imprecision 6.2%) as covariates on clearance. All three model(s) facilitated accurate and precise estimation of vancomycin AUC in critically ill children.

## 1. Introduction

Vancomycin is the drug of choice for treatment of serious Gram-positive infections in children and is one of the most frequently administered drugs in the pediatric intensive care unit [1]. Its efficacy and toxicity are most closely related to an individual’s 24 h area under the curve (AUC_24_) [2,3,4,5,6]. Traditionally, vancomycin therapeutic drug monitoring (TDM) relied upon measurement of trough concentrations (C_min_), which were used as a surrogate for AUC_24_ [7]. However, with the continued maturation of Bayesian dosing software programs, AUC-based dosing using population models and limited sampling is becoming more routine [8]. Unfortunately, Bayesian AUC estimation from a single trough measurement can be inaccurate, and validated vancomycin population pharmacokinetic (popPK) models to inform Bayesian dosing and AUC estimation in critically ill children are lacking.

Vancomycin is renally eliminated, and total body clearance (CL) is correlated with glomerular filtration rate (GFR) [9]. However, direct measurement of GFR in critically ill children is impractical and biomarkers are typically used to estimate renal function in individual patients. Creatinine is the biomarker most often relied upon to estimate kidney function in children, but it is not ideal in critically ill children as serum concentrations are affected by numerous factors (medications, muscle mass, age) and are slow to change in the setting of acute kidney injury (AKI) [10]. Despite their limitations, creatinine-based GFR equations, such as the bedside Schwartz equation (Schwartz) [11], remain commonly used for dosing guidance in the critical care setting.

Newer biomarkers have been discovered that are more sensitive indicators of kidney injury and function than creatinine. Cystatin C (CysC) is a protein that is widely expressed by nucleated cells, produced at a constant rate in the body, freely filtered by the glomerulus, and not secreted by renal tubules [12]. These characteristics make plasma CysC a good biomarker of GFR [12] and studies have demonstrated its superiority over creatinine for GFR estimation and earlier AKI detection in critically ill children, including those with sepsis [13,14,15,16,17,18,19]. We also previously found that CysC-based eGFR was more closely associated with vancomycin clearance (CL) than creatinine-based eGFR in critically ill children using a popPK modeling approach [20]. Neutrophil gelatinase-associated lipocalin (NGAL) is another promising biomarker for the early detection of AKI in critically ill children [19,21,22]. Plasma and urinary NGAL increase prior to changes in creatinine in critically ill children with sepsis [19,23], and urinary NGAL concentrations have been described as a predictor of vancomycin-associated AKI in hospitalized adults [24]. Other urinary biomarkers, kidney injury molecule-1 (KIM-1) and osteopontin, show good correlation with vancomycin exposures (AUC_24_, C_max_) and are predictive of vancomycin-associated AKI in humanized rat models [25,26,27]. Given the known limitations of creatinine in children, we hypothesized that the clinical use of novel biomarkers can improve estimation of kidney function and vancomycin clearance in critically ill children and ultimately promote individualized vancomycin dosing via Bayesian approaches.

We aimed to develop a popPK model for intravenous (IV) vancomycin in critically ill children incorporating novel urinary and plasma biomarkers of kidney injury. From this population model, we sought to evaluate Bayesian estimation of AUC_24_ in a separate cohort of patients. Ultimately, the goal of this work was to generate and validate a population PK model of vancomycin in critically ill children that could inform Bayesian estimation of AUC_24_ using limited sampling.

## 2. Materials and Methods

### 2.1. Study Population

We performed a prospective, observational study in the Pediatric Intensive Care Unit (PICU) at the Children’s Hospital of Philadelphia (CHOP) from August 2018 to July 2021. Patients aged 1–17 years old receiving intermittent dosing of intravenous (IV) vancomycin for a suspected infection, defined as performance of a microbiological culture within 24 h of vancomycin initiation, were eligible for inclusion. Eligible patients were identified as soon after initiation of vancomycin as possible. Those receiving renal replacement therapy, plasmapheresis, or extracorporeal membrane oxygenation were ineligible. To be deemed evaluable, subjects had to have ≥3 PK samples collected, as well as ≥1 urine sample and 1 plasma sample collected for biomarker measurement (see below).

After enrollment was completed, evaluable subjects were divided into model training (n = 30) and testing (n = 20) groups. To perform this, we randomly selected 20 subjects who had ≥4 PK samples obtained to serve as the model testing group, while the remainder were assigned to the model training group. This approach was taken to facilitate AUC_24_ estimation using noncompartmental methods within the model testing group, as described below.

The CHOP Institutional Review Board (IRB) approved the study protocol (IRB 18-014851, Approved: 20 March 2018) with a waiver of documented assent; verbal assent was obtained, as appropriate. Documented informed consent was obtained from the patient’s parent(s)/legal guardian(s).

### 2.2. Dosing, PK Sampling, and Biomarker Measurement

Vancomycin was ordered for clinical care in all patients, with dosages and infusion rates determined by the clinical team. Typical initial dosages were 10–15 mg/kg/dose every 6–8 h, depending on age, weight, and estimated renal function. All decisions about dosing and duration of therapy were at the discretion of the clinical team. During the study, routine TDM included collection of trough (C_min_) concentrations, with doses adjusted to achieve a goal of 5–15 µg/mL, as appropriate. In 2020, AUC-based TDM was implemented at our hospital, such that patients requiring >48 h of vancomycin therapy had two blood samples collected (at 1 h after the end of the infusion and at C_min_). AUC was calculated using log-linear methods and doses were adjusted to achieve an AUC of 400–600 mg h/L.

For each participant, five vancomycin concentrations (PK sampling) were obtained after ≥4 vancomycin doses during a single dosing interval at the goal times shown in Appendix A; since clinical care could interfere with the precise timing of sample collection, samples were accepted outside of these windows. Sampling took place within 48 h of enrollment for all participants. Samples could be collected prior to the fourth dose if a patient had impaired renal function such that he/she did not receive vancomycin at regular dosing intervals, so long as he/she had received ≥24 h of treatment; this applied to only one participant. All samples were collected via arterial catheter, peripheral venipuncture, or venous catheter, if not used for administration of vancomycin. In addition to PK sampling, the clinical team performed therapeutic drug monitoring (TDM) according to institutional standard practice. Results of TDM samples were recorded and included in our study if collected prior to PK sampling; individuals only participated through the time of PK sampling, so dosing information and TDM results that took place after PK sampling for the study were not recorded.

All vancomycin concentrations were measured by chemiluminescent microparticle immunoassay (Abbott, Abbott Park, IL, USA) in the CHOP Chemistry Laboratory; the lower limit of quantification (LLOQ) of this assay was 3.0 µg/mL. Results of both PK sampling and TDM were made available in the subject’s electronic medical record.

After enrollment, urine and blood samples were obtained for biomarker measurement. Two urine samples were collected from subjects for measurement of the following biomarkers: NGAL, KIM-1, cystatin C, osteopontin, and creatinine. These samples were obtained starting the day of enrollment in the evening (3–7 pm) and morning (6–10 am) prior to PK sampling. Collection of urine samples outside of the above windows was permitted, based on the condition and clinical care needs of the patient. Urine samples were collected via indwelling urinary catheter, clean intermittent catheterization (if performed for clinical care), cotton balls, urine cup, or urine bag. In addition to prospective urine collection, we also identified and obtained any available residual urine samples that had been collected clinically within 24 h prior to initiation of IV vancomycin. These samples served as baseline biomarker measurements when available. A single blood sample was drawn within 24 h of PK sampling for measurement of plasma biomarkers (cystatin C and NGAL), as well as creatinine if not performed clinically. We similarly identified any available residual plasma samples that had been collected clinically up to 24 h preceding initiation of IV vancomycin to serve as baseline measurement of biomarkers.

KIM-1, osteopontin, NGAL (urine and plasma), and cystatin C (urine and plasma) were measured using Quantikine^®^ ELISA (R & D Systems, Inc., Minneapolis, MN, USA). Urine creatinine (uCr) was measured via two-point end enzymatic method on a Cobas^®^ system (Roche Diagnostics, Basel, Switzerland). These tests were performed in the CHOP Translational Core Laboratory. Plasma creatinine was measured by two-point rate spectrophotometric method (Vitros5600™ analyzer, Ortho Clinical Diagnostics, Raritan, NJ, USA) in the CHOP Chemistry Laboratory.

### 2.3. Population PK Model Training

Figure 1 displays a flow diagram of our approach to population PK model training and testing. Nonparametric population PK modeling was performed using the Pmetrics package (version 1.9.7; Laboratory of Applied Pharmacokinetics and Bioinformatics, Los Angeles, CA, USA) [28] for R (version 3.6.3; R Foundation for Statistical Computing, Vienna, Austria) [29] in RStudio (v1.2.5033; RStudio, Inc., Boston, MA, USA) [30]. One- and two-compartment models were constructed using the nonparametric adaptive grid (NPAG) algorithm [31]. Parameters included V_d_ (volume of the distribution) and CL (total body clearance) for one-compartment models and CL (clearance), Q (intercompartmental clearance), V1 (central volume), and V2 (peripheral volume) for two-compartment models. Clearance parameters were allometrically scaled for standardized weight to a power of 0.75 and volume parameters were scaled by standardized weight (power 1); weight was standardized by the median of the subjects’ weights (27 kg). The weighting function on observations was 1/(gamma * SD^2^), where SD (standard deviation) was a combined additive and multiplicative function of the assay imprecision as a polynomial equation: SD = C_0_ + [C_1_ × observed concentration]; C_0_ had a value of 1.5 (half the LLOQ) and C_1_ a value of 0.1, i.e., an assay with 10% coefficient of variation (CV%). Gamma was initially set to 1 and fitted to estimate the residual model error.

Following determination of the base model structure, covariate model selection was then performed in a two-stage process. Since vancomycin is renally eliminated, we first sought to identify the best renal function marker to include as a covariate on CL. This included plasma biomarkers (creatinine, cystatin C (pCysC), and NGAL (pNGAL)), as well as eGFR based on these biomarkers. GFR was estimated using creatinine alone according to the bedside Schwartz equation (Schwartz_bed_) [11], pCysC alone based on the Hoek equation [32], and both creatinine and pCysC based on the full age spectrum equation [33]. These cystatin C equations were chosen based on our previous evaluation of their ability to inform vancomycin CL during parametric population PK modeling [20].

Next, we evaluated additional covariates on CL and V1 using a forward selection approach. All covariates were selected based on physiologic plausibility. Binary covariates included sex, receipt of vasopressor medications, and presence of augmented renal clearance (defined as eGFR > 130 mL/min/1.73 m^2^; tested on CL only) [34], while continuous covariates included age, Pediatric Index of Mortality 3 (PIM3) score [35], and the above listed plasma and urinary biomarkers (on CL only). Urinary biomarkers were normalized to urine creatinine (i.e., [biomarker]/[uCr]) to account for urine volume. Continuous covariates were normalized to the median population value with the covariate effect evaluated using a power function; age was also evaluated as a Hill function on CL [36]. Meanwhile, binary covariates were parameterized using a linear proportional approach such that the covariate effect reflects a proportion increase/decrease of the typical parameter value in the absence of the covariate.

Covariate selection was guided by the principle of parsimony and by measures of goodness-of-fit. Models were evaluated at each step by inspection of observed-versus-predicted concentration plots, as well as examination of the model’s bias, imprecision, regression coefficient, log-likelihood ratio (−2 × LL), and Akaike’s information criterion (AIC) value. With the addition of a covariate to a model, a difference in the log-likelihood ratio of >3.84 was considered significantly improved fit (corresponding to *p* < 0.05 for one additional degree of freedom). When comparing models with the same degrees of freedom, lower AIC, bias, and imprecision of the observed-versus-predicted concentrations guided model selection.

### 2.4. Model Testing via Area under the Curve Comparisons

After model training, we evaluated the ability of the full model to predict vancomycin AUC for each subject in the model testing group. We first calculated the AUC for the dosing interval during which PK sampling was performed using his/her observed concentrations and linear up-log down trapezoidal noncompartmental analysis (NCA) [37]. Because of the timing of PK sampling, the drug was assumed to be at steady state, and therefore the same minimum concentration (C_min_) was used prior to and after a given dose in these calculations. The 24 h “true” or observed AUC (AUC_obs_) was then computed based on the subject’s anticipated number of doses in a 24 h period.

We then used Bayesian estimation to evaluate how well our full model predicted AUC_obs_. The population joint density of the full model was employed as a Bayesian prior for each subject in the model testing group. Simulated concentration–time profiles were generated with predicted outputs each minute. The predicted concentration at the time of each measured PK sample, as well as at the start and end of his/her infusion, was recorded. The trapezoidal method was then taken, using all of the subject’s predicted concentrations to calculate a predicted AUC_24_ (AUC_full_).

Recognizing that urinary biomarkers would not be routinely available in clinical practice, we also performed Bayesian estimation using two simpler models: a model incorporating only CysC-based eGFR (from Hoek equation) on CL and a model incorporating only creatinine-based eGFR (from bedside Schwartz equation) on CL. These models were chosen as comparators because they are more parsimonious and contain covariates that are readily available at all (creatinine) or many (CysC) pediatric institutions, which we felt would allow for easier clinical implementation. As above, AUC_24_ was calculated for both the Hoek (AUC_Hoek_) and Schwartz (AUC_Schwartz_) models.

To assess the predictive performance of our models for AUC estimation, we determined the bias and imprecision of the AUC predictions from each model for each subject. Bias was calculated as (AUC_pred_ − AUC_obs_)/AUC_obs_ × 100 and imprecision as |AUC_pred_ − AUC_obs_|/AUC_obs_ × 100, where AUC_pred_ is the generic AUC_24_ predicted by our models: AUC_full_, AUC_Hoek_, and AUC_Schwartz_. Data were then summarized as the median bias (median percentage predictive error) and imprecision (median absolute percentage predictive error) for each model, as well as the fraction of subjects whose AUC_pred_ was within 5%, 10%, 15%, and 20% of his/her AUC_obs_. Spearman correlation between AUC_pred_ and AUC_obs_ was also determined for each model. The a priori acceptance criteria were that AUC_pred_ was within 20% of AUC_obs_ in 85% of subjects and that the correlation between AUC_pred_ and AUC_obs_ was >0.9.

### 2.5. Area under the Curve Estimation from Optimal Sampling

The goal of Bayesian AUC estimation in clinical practice is to accurately estimate vancomycin AUC_24_ using limited PK sampling. As such, we also sought to evaluate the ability of our model(s) to estimate AUC_24_ using one and two optimally timed PK samples per subject. To achieve this, we utilized the multiple-model optimization algorithm (MMopt) in Pmetrics, which finds the optimal times based on when all the PK curves generated by the support points in the nonparametric model are most separated (e.g., the time points that are most informative), minimizing the Bayesian risk of misclassifying an individual as the wrong set of support points [38]. For each subject in the model testing group, MMopt was used identify the one and two most informative sampling time points for Bayesian estimation of AUC. We then created a reduced dataset for each subject that contained only the one or two observed concentrations closest to the optimal sampling time(s) identified by MMopt. We repeated the processes above in 2.d. with the population joint density of each model employed as a Bayesian prior and simulated concentration–time profiles generated using only these reduced datasets. Predicted concentrations were again recorded and AUC_24_ was calculated via the trapezoidal method. Bias, imprecision, and correlation were determined for each model, this time based on predictions from limited sampling.

## 3. Results

### 3.1. Study Population

In total, 85 patients provided consent to participate. Five were deemed screen failures following consent and were excluded prior to performance of any study procedures. Meanwhile, 29 subjects were unevaluable because vancomycin was discontinued by the clinical team prior to PK sampling (n = 28) or could not undergo PK sampling (n = 1). One additional subject was excluded due to laboratory processing issues of the PK samples, making results uninterpretable (results reported out of order). As a result, 50 individuals participated in our study and were fully evaluable. Of these, 44 were eligible to be in the testing group (i.e., had ≥4 PK samples available), of whom we randomly chose 20. The characteristics of the model training and test groups are shown in Table 1. In general, the groups were similar, although the test group were less often on vasopressors at the start of vancomycin and received slightly larger vancomycin doses at the time of PK sampling.

### 3.2. Population PK Model Training

Thirty subjects contributed 150 vancomycin concentrations (14 clinical samples and 136 research PK samples) towards model training. The range of vancomycin concentrations was 3.9 to 67.8 µg/mL. Two concentrations were below the limit of quantification and coded as LLOQ/2 (i.e., 1.5 µg/mL). Observed concentrations vs. time after dose are plotted in Appendix A.

Appendix A displays the model training steps. A two-compartment model best described the data (AIC 820.5 vs. 879.6 for one-compartment). When evaluating renal function covariates on vancomycin CL, cystatin C outperformed SCr (AIC difference of −14.6) and pNGAL (AIC difference of −14.3). Meanwhile, eGFR based on the Hoek equation had a lower AIC (793.3) compared to eGFR based on the Schwartz equation (807.6) or the full age spectrum equation (801.5). Incorporation of eGFR_Hoek_ or pCysC on CL resulted in comparable −2 × LL, AIC, and population predicted vs. observed R^2^ values. We proceeded with further model training using the eGFR_Hoek_ model since clinical dosing guidance is typically based on a patient’s eGFR rather than a direct biomarker result.

When evaluating the addition of other covariates on CL and V1, numerous covariates, including urinary biomarkers on CL, improved model fitness based on a reduction in −2 × LL and AIC (Appendix A). However, the incorporation of the uNGAL on CL, parameterized as the natural logarithm of uNGAL/uCr, led to the largest −2 × LL and AIC reductions. The correlation between log uNGAL and eGFR_Hoek_ was low (−0.27). No additional covariates were informative on vancomycin CL or V1; thus, this model constituted the full model for testing. Observed versus population and individual predicted concentration plots of the full model are shown in the Appendix A, and the population PK parameter estimates from this model are shown in Table 2.

### 3.3. Area under the Curve Comparisons

Twenty subjects comprised the model testing group. These subjects were similar to the model training group in terms of age, weight, PIM3 scores at PK sampling, timing of PK sampling, and vancomycin dosages received (Table 1), although they less often received vasopressors and had higher eGFR at vancomycin initiation. Biomarker concentrations at the time of PK sampling were also similar between the two groups (Appendix A).

The median AUC_24_ among the 20 subjects calculated using the trapezoidal method and observed concentrations (AUC_obs_) was 456 mg h/L (range: 367–885). When using estimated concentrations from the individual Bayesian posteriors and our full model (AUC_full_), the median AUC_24_ was 475 mg h/L (range: 327–907). The median bias and imprecision of AUC_full_ compared to AUC_obs_ were 2.3% and 6.2%, respectively, and the correlation between the AUC estimates was 0.939. Meanwhile, 90% of subjects’ AUC_full_ estimates were with 20% of their AUC_obs_.

The PK parameter estimates for the two simplified models (based on eGFR_Hoek_ and eGFR_Schwartz_) are shown in Appendix A. The median bias and imprecision of AUC_Hoek_ compared to AUC_obs_ were 1.8% and 7.0%, respectively, and the correlation between AUC_Hoek_ and AUC_obs_ was 0.926. The median bias and imprecision of AUC_Schwartz_ compared to AUC_obs_ were -2.4% and 6.2%, respectively, and the correlation between AUC_Schwartz_ and AUC_obs_ was 0.893. Additionally, 85% and 95% of subjects’ AUC estimates were with 20% of their AUC_obs_ using the Hoek and Schwartz models, respectively.

Observed versus individual predicted concentrations (i.e., Bayesian posteriors) for each of the three models are shown in Figure 2, while Table 3 displays the performance parameters of these models to estimate AUC_24_. The optimal sampling times relative to the end of the infusion for each of the three models are shown in Appendix A. When fitting all available PK samples, the three models performed similarly (with AUC_obs_ as the reference). However, when fitting the one or two PK samples closest to the MMopt optimal times, the performance of our full model declined. The imprecision of AUC_full_ more than doubled when using one or two PK samples, and fewer subjects’ AUC_24_ were within 20% of AUC_obs_. However, ≥85% of subjects’ AUC_24_ from the Hoek and Schwartz models were within 20% of AUC_obs_ when using limited sampling.

## 4. Discussion

In this population PK study, cystatin C was superior to the traditional renal function biomarker of serum creatinine as a marker of vancomycin clearance in critically ill children. This is consistent with other studies that described better correlation between vancomycin CL and CysC-based eGFR than creatinine-based eGFR in noncritically ill and critically ill pediatric patients [20,39]. Contrary to our previous work [20], however, we found that eGFR based on CysC alone (using the Hoek equation) outperformed eGFR based on both CysC and creatinine (full age spectrum equation). This highlights the inadequacy of creatinine as a renal function marker in critically ill children and suggests that routine measurement of CysC when administering vancomycin may be a more reliable approach. As the availability of cystatin C at pediatric institutions increases, and experience with cystatin C-based eGFR equations mounts, cystatin C may replace creatinine as the biomarker to inform dosing of other renally eliminated drugs in pediatrics as well. With increased appreciation of the negative ramifications of AKI on clinical outcomes, it is crucial to provide safe doses of nephrotoxic medications such as vancomycin. Given the well-recognized limitations of creatinine in pediatric patients, particularly in the ICU setting, it may be time to move towards a better biomarker in our most vulnerable children.

A goal of this work was to explore the potential value of novel urinary biomarkers to describe vancomycin disposition in critically ill children using a popPK modeling approach. Since changes in serum creatinine and other blood markers of kidney function may be delayed in the setting of AKI [40], we hypothesized that urinary biomarkers could facilitate detection of fluctuations in vancomycin CL before blood biomarkers. In fact, urinary NGAL was an influential covariate on vancomycin CL during our full model training; other urinary biomarkers (KIM-1, CysC, osteopontin) also led to large reductions in the AIC during model training steps, although not to the same extent as NGAL. We believe that these findings are important and warrant further investigation. Rather than solely relying on blood biomarkers to signify kidney function, bedside measurement of urinary biomarkers could provide insight into which patients have subclinical kidney injury (i.e., not detected via blood biomarkers) and may require early, pre-emptive dose adjustments. Although biomarkers did not improve estimation of AUC_24_ estimation in our testing group, they could potentially be used as screening tests to identify patients at highest risk for impaired kidney function and/or toxicity.

Urinary biomarkers can detect subclinical kidney injury, but the majority of patients in our study actually had augmented renal clearance (ARC). This phenomenon describes a state of hyperfiltration and increased drug clearance, which can be detrimental in critically ill patients with severe infections, and it is unclear how reliable urinary biomarkers are in that clinical situation. Although the precise definition of ARC in children has been debated, an eGFR >130 mL/min/1.73 m^2^ has been utilized in other vancomycin studies [41]. In our study, half of each of the model training and testing groups had ARC by Hoek equation calculation. This may explain why our full model failed to outperform the Schwartz and Hoek model in terms of AUC_24_ estimation, as urinary biomarkers may not perform as well in patients with ARC as they do in those with AKI. Similarly, eGFR equations are generally more accurate in patients with impaired renal function rather than in ARC. Given the small sample size of our model testing group and limited number of subjects with impaired renal function, we may have been unable to fully demonstrate the value of novel biomarkers (both urinary and plasma) when it comes to AUC estimation.

It is also possible that urinary NGAL measurement is not precise enough to substantially influence AUC_24_ estimation, particularly when using limited PK sampling. Urinary NGAL ranged from 1.4 to 2809 ng/mg creatinine in the model training group and from 7.8 to 10,034 ng/mg creatinine in the model testing group. Although measurements were not significantly different between the groups using Wilcoxon rank sum testing, this large variability in urinary biomarker values may preclude precise AUC estimation at an individual level using Bayesian methods. While we did not explore utilizing cut-points to categorize biomarker values (e.g., low, medium, high values), that is a potential avenue for future investigation.

Another goal of this study was to develop a popPK model that could be used to inform AUC-based vancomycin dosing using Bayesian estimation and limited sampling. We utilized a nonparametric popPK modeling approach, which differs from the more widely used parametric popPK methods in that it assumes that the distribution of parameter values is not necessarily described by an a priori defined continuous function (e.g., normal distribution) [42]. As a result, more commonly used statistical measures of variability in parametric popPK models (e.g., mean, standard deviation, coefficient of variation) may not fully describe the shape and structure of a nonparametric distribution and, as such, the idea of typical parameter values and interindividual variability around them does not apply. However, because nonparametric approaches allow for parameter probability distributions to take any shape, subpopulations and outliers can be detected, which is ideal in a popPK study of a highly dynamic population such as critically ill children, and methods for Bayesian estimation of individual PK parameters are robust. A more thorough review of nonparametric and parametric popPK approaches has been published [43].

At the time that this study started, AUC estimation was not standard of care. However, AUC-based dosing is now routine at our institution for anyone receiving >48 h of treatment. Clinical pharmacists implement log-linear methods to calculate AUC based on two vancomycin concentrations. To provide a distinct advantage over this approach, we felt it was important to specifically evaluate the utility of our models to inform AUC estimation based on a single, optimally timed vancomycin measurement. Both the Hoek and Schwartz models accurately estimated AUC_24_ (within 20% of AUC_NCA_) in 85% of subjects from a single sample with a median bias of 2.5% and −0.9%, respectively. An ongoing, prospective observational study (NCT05691309) at our institution is evaluating how well Bayesian AUC using our Hoek model aligns with clinical AUC calculations (using log-linear calculations) and assessing the ways in which this would influence dosing recommendations.

There are limitations to our study. First, we did not enroll children younger than one year of age and so our model(s) cannot be applied to infants. Cystatin C values are affected by age in this group, likely due to renal maturation occurring over the first year of life; thus, infants were specifically excluded from our study. A recent popPK study in critically ill neonates found that a model including creatinine on vancomycin CL performed similarly to a model including cystatin C and age [43]; thus, cystatin C may not be a superior renal function biomarker in the neonatal population. Second, because we required that a subject have at least four PK samples to be included in the model testing group, it is possible that our process of group assignment introduced bias; hence, the groups were different. We did not detect any statistically significant differences in the blood or urinary biomarkers between these groups, which were the covariates included in our models, but it is possible that other factors not considered covariates during model development in the training group were influential on vancomycin PK and AUC_24_ estimation within the smaller model testing group. Third, recently published data suggest that urine chemistry analytes can be affected by collection of urine using cotton balls [44]. We are unaware of studies that have evaluated this for the urinary biomarkers included in our study, but this should be explored in future studies as urine collection methods could be a potential source of variability that affect the association between biomarkers and vancomycin clearance. Lastly, urinary biomarkers are sensitive for detection of AKI and, thus, may fluctuate over time. Although we collected ≥2 samples per subject, it is possible that changes in biomarker concentrations occurred outside of the windows of our urine sample collection. Although understanding the fluctuations of urinary biomarkers in critically ill children would be of interest, the need to collect urine at very specific times or to collect samples more often would further limit the clinical applicability of these biomarkers to inform drug dosing.

## 5. Conclusions

The present study developed a two-compartment model to describe vancomycin PK in critically ill children. Plasma cystatin C and urinary NGAL were informative on vancomycin clearance in our full popPK model. However, the full popPK model was not superior for estimation of Bayesian AUC_24_ compared to simpler popPK models that included only cystatin C- or creatinine-based eGFR on clearance. Future studies will evaluate the utility of our models to inform vancomycin dosing using Bayesian estimation compared to two-point log-linear regression calculations.

## Figures and Tables

**Figure 1 pharmaceutics-15-01336-f001:**
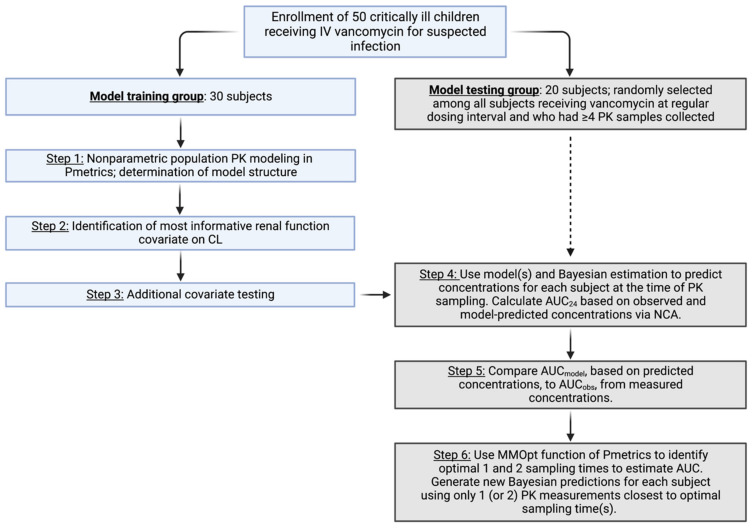
Flow diagram of study design. **Left**-hand side depicts how data from the 30 subjects assigned to model training were used, while **right**-hand side shows how data from the 20 subjects in the model testing group were utilized. Abbreviations: AUC, area under the concentration–time curve; AUC_model_, area under the curve from model-predicted concentrations; AUC_obs_, area under the curve from observed concentrations; CL, clearance; MMOpt, multiple-model optimization algorithm; NCA, noncompartmental analysis; PK, pharmacokinetic. Created with BioRender.com.

**Figure 2 pharmaceutics-15-01336-f002:**
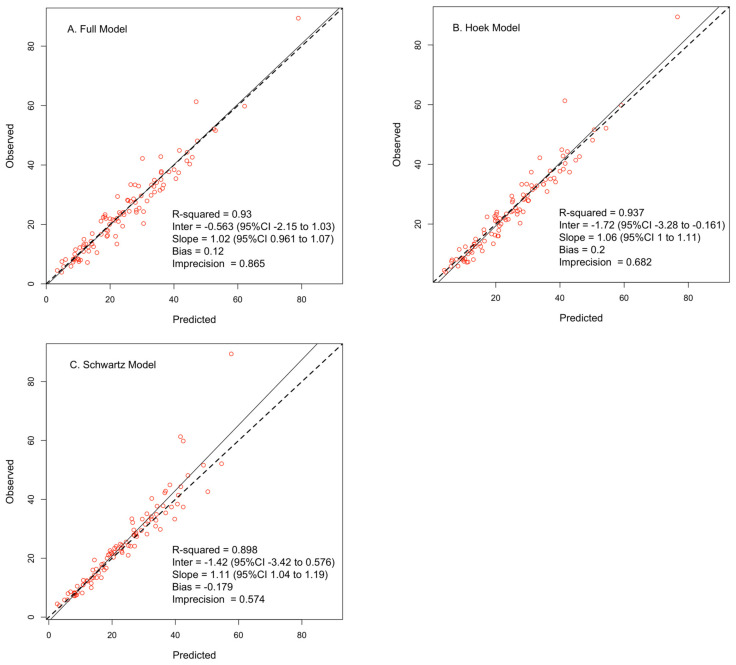
Observed versus posterior (individual) predicted concentrations for the full (**A**), Hoek (**B**), and Schwartz (**C**) models.

**Table 1 pharmaceutics-15-01336-t001:** Study population characteristics.

Characteristic	Model Training Group (n = 30)	Model Testing Group (n = 20)	*p*-Value ^a^
**At start of vancomycin**			
Age in years, median (IQR)	9.8 (3.8–11.2)	10.2 (2.9–13.2)	0.68
Weight in kg, median (IQR)	25.9 (13.9–41.8)	37.6 (14.4–57.5)	0.33
Female sex, n (%)	11 (37)	5 (25)	0.54
Serum creatinine in mg/dL, median (IQR)	0.40 (0.23–0.60)	0.40 (0.19–0.43)	0.22
eGFR_Schwartz_ in mL/min/1.73 m^2^, median (IQR)	117 (97–154)	154 (126–185)	0.06
Receipt of vasopressors, n (%)	18 (60)	6 (30)	0.05
Vancomycin dose in mg/kg/dose, median (IQR)	14.7 (11.9–15.0)	14.5 (11.8–14.8)	0.60
**At PK sampling**			
Serum creatinine in mg/dL, median (IQR)	0.30 (0.20–0.48)	0.35 (0.19–0.50)	0.92
eGFR_Schwartz_ in mL/min/1.73 m^2^, median (IQR)	164 (114–222)	156 (134–184)	0.95
eGFR_Hoek_ in mL/min/1.73 m^2^, median (IQR)	143 (110–197)	130 (96–156)	0.64
Receipt of vasopressors, n (%)	14 (47)	7 (35)	0.56
PIM3 probability of death, median (IQR)	1.3% (0.5–4.3)	1.3% (0.4–4.1)	0.96
Vancomycin dose in mg/kg/dose, median (IQR)	13.2 (10.0–14.8)	15.0 (14.5–15.7)	0.005
Duration of vancomycin therapy prior to PK sampling (in hours), median (IQR)	36.4 (30.8–41.4)	36.1 (32.1–41.9)	0.68

^a^ Continuous variables were compared using Wilcoxon rank sum tests and categorical variables were compared using chi-squared or Fisher’s exact tests. Abbreviations: eGFR, estimated glomerular filtration rate; IQR, interquartile range; PIM3, Pediatric Index of Mortality 3.

**Table 2 pharmaceutics-15-01336-t002:** Population PK parameter estimates for the full PK model.

Parameter	Weighted Parameter Estimate		CV%	Shrinkage %
	Median	95th Percentile		
CL0	3.31	2.53–4.22	39.5	54.7
CL_WT_	0.75	-	-	
CL_HOEK_	0.85	0.22–0.90	62.8	55.6
CL_NGAL_	0.94	0.86–1.00	10.9	50.5
V_C_0	3.50	2.72–7.09	49.2	59.5
V_C-WT_	1	-	-	
Q0	7.09	4.76–7.97	32.6	60.9
Q_WT_	0.75	-	-	
V_P_0	7.75	6.63–13.80	39.0	49.1
V_P-WT_	1	-	-	

Full model parameterized as: CL = CL0 · (WT/27)^CLWT^ · (HOEK/134)^CLHOEK^ · (CL_NGAL_)^LNGAL^, V1 = V_C_0 · (WT/27)^VCWT^, Q = Q0 · (WT/27)^QWT^, V2 = V_P_0 · (WT/27)^VPWT^. Hoek is the estimated GFR based on the cystatin C-based Hoek equation; LNGAL is the natural logarithm of the urinary NGAL concentration normalized to urinary creatinine (uNGAL/uCr). Abbreviations: CL, clearance; Q, intercompartmental clearance; V1, central volume; V2, peripheral volume.

**Table 3 pharmaceutics-15-01336-t003:** Performance of models to estimate AUC_24_ via Bayesian estimation (AUC_model_) compared to AUC calculated using observed concentrations (AUC_obs_) ^a^.

	Full Model	Hoek Model	Schwartz Model
All available PK samples			
AUC_24_, median (range)	475 (325–857)	488 (338–907)	501 (319–710)
Median bias ^b^	2.3%	1.8%	−2.4%
Median imprecision ^c^	6.2%	7.0%	6.2%
Number of subjects with AUC_model_ within 20% of AUC_obs_	18 (90%)	17 (85%)	19 (95%)
Correlation between AUC_model_ and AUC_obs_	0.939	0.926	0.893
Two optimally timed PK samples			
AUC_24_, median (range)	487 (316–968)	482 (333–832)	471 (332–714)
Median bias ^b^	2.0%	1.8%	−1.2%
Median imprecision ^c^	13.4%	6.0%	6.8%
Number of subjects with AUC_model_ within 20% of AUC_obs_	15 (75%)	18 (90%)	17 (85%)
Correlation between AUC_model_ and AUC_obs_	0.876	0.928	0.860
Single optimally timed PK sample			
AUC_24_, median (range)	526 (284–968)	523 (283–830)	466 (353–700)
Median bias ^b^	3.7%	2.5%	−0.9%
Median imprecision ^c^	13.4%	11.4%	7.0%
Number of subjects with AUC_model_ within 20% of AUC_obs_	13 (65%)	17 (85%)	17 (85%)
Correlation between AUC_model_ and AUC_obs_	0.817	0.891	0.825

^a^ All model-derived AUC estimates were compared to AUC calculated using the trapezoidal methods on observed concentrations (AUC_obs_); median AUC_obs_: 456 mg h/L, range: 367–885 mg h/L. ^b^ Bias calculated as: (AUC_model_ − AUC_obs_)/AUC_obs_ × 100. ^c^ Imprecision calculated as: |AUC_model_ − AUC_obs_|/AUC_obs_ × 100.

## Data Availability

Data can be made available upon request.

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
