# Peer review of "Optimizing Vancomycin Therapy in Critically Ill Children: A Population Pharmacokinetics Study to Inform Vancomycin Area under the Curve Estimation Using Novel Biomarkers"

_pharmaceutics, 2023, doi:10.3390/pharmaceutics15051336_

Round 1

Reviewer 1 Report

The manuscript entitled " Optimizing Vancomycin Therapy in Critically Ill Children: A Population Pharmacokinetics Study to Inform Vancomycin Area-Under-the-Curve Estimation Using Novel Biomarkers" by Downes KJ et al presents a novel pharmacokinetic model of vancomycin in critically ill pediatric patients. The manuscript is well written, methods are exhaustive, results are presented clearly, and the discussion is in accordance with the results obtained by the authors. They propose that serum creatinine may not be the best biomarker for vancomycin clearance and instead propose the use of cystatin c as an important covariate in the PoPK model for this population. Some comments that may improve the manuscript:

1- Could the authors better explain the following statement?  "Samples could be collected prior to the fourth dose if a subject’s renal function required administration of vancomycin at >12-hour intervals and the subject had received ≥24 hours of treatment."  

2- Have the authors assessed beforehand that urine samples collected with cotton balls gave similar results as with other sampling methods? PLease check the following article. DOI: 10.1016/j.jpeds.2022.02.051

 3- Could the authors discuss further why Bayesian AUC could be better explained using a model that only includes creatinine levels for eGFR calculation when this covariate is not retained in the final model?

Author Response

Thank you for the thoughtful comments and critiques, which have greatly strengthened the clarity and the overall message of our work. We have responded to this feedback and provide a point-by-point description of the changes we have made below. Specific changes are highlighted with italics below and in the tracked changes version of the manuscript, which accompanies this submission.

Point 1. Could the authors better explain the following statement?  "Samples could be collected prior to the fourth dose if a subject’s renal function required administration of vancomycin at >12-hour intervals and the subject had received ≥24 hours of treatment."  

  • Thank you for your critique. This was meant to state that subjects who had impaired renal function and therefore were not on a routine dosing schedule (e.g. had trough-guided dosing) could also be included. This applied to only 1 participant. We have tried to clarify this statement by rephrasing it as follows, “Samples could be collected prior to the fourth dose if a patient had impaired renal function such that he/she did not receive vancomycin at regular dosing intervals, so long as he/she had received ≥24 hours of treatment; this applied to only one participant."  

Point 2. Have the authors assessed beforehand that urine samples collected with cotton balls gave similar results as with other sampling methods? Please check the following article. DOI: 10.1016/j.jpeds.2022.02.051

  • Thank you for this suggestion. We followed urine collection protocols implemented in studies examining biomarkers for detection of acute kidney injury in children but did not evaluate whether different collection methods could affect urinary biomarker concentrations. Given this possibility, we have added this as a potential limitation in the Discussion:
    Third, recently published data suggest that urine chemistry analytes can be affected by collection of urine using cotton balls.44 We are unaware of studies that have evaluated this for the urinary biomarkers included in our study. But, this should be explored in future studies as urine collection methods could be a potential source of variability that affects the association between biomarkers and vancomycin clearance.”

Point 3. Could the authors discuss further why Bayesian AUC could be better explained using a model that only includes creatinine levels for eGFR calculation when this covariate is not retained in the final model?

  • We apologize if this was not clear. Commentary on this is included in the Discussion. In terms of creatinine vs. cystatin c, both the Schwartz and Hoek models demonstrated good performance in estimating AUC, even with limited sampling. Thus, we believe that our full model performed less well due to variability in the NGAL concentrations between the model training and testing groups, differences in renal function of participants in the two groups, or possibly a product of the small sample size of the testing group. Further, the benefits of accurate GFR estimation and use of renal injury biomarkers are most pronounced in patients with impaired renal function, where creatinine is very limited. Unfortunately, we had fewer patients with low versus high eGFR. Thus, the ability of our full model to improve AUC estimation may have been impacted by the population studied/included. We have elaborated on this in our Discussion:
  • “Urinary biomarkers can detect subclinical kidney injury, but the majority of patients in our study actually had augmented renal clearance (ARC). This phenomenon describes a state of hyperfiltration and increased drug clearance, which can be detrimental in critically ill patients with severe infections, and it is unclear how reliable urinary biomarkers are in that clinical situation. Although the precise definition of ARC in children has been debated, an eGFR >130 mL/min/1.73 m2 has been utilized in other vancomycin studies.41 In our study, half of each of the model training and testing groups had ARC by Hoek equation calculation. This may explain why our full model failed to outperform the Schwartz and Hoek model in terms of AUC24 estimation, as urinary biomarkers may not perform as well in patients with ARC as they do in those with AKI. Similarly, eGFR equations are generally more accurate in patients with impaired renal function rather than in ARC. Given the small sample size of our model testing group and limited number of subjects with impaired renal function, we may have been unable to fully demonstrate the value of novel biomarkers (both urinary and plasma) when it comes to AUC estimation.”

Reviewer 2 Report

This is a well-written high-quality manuscript by Downes et al. on the development of a popPK model from critically ill children to estimate AUC values based on novel biomarkers of renal clearance, in particular cystatin-C. The clinical rationale for the study is clear, methods appropriate, interpretation of results justified, and the discussion, including some limitations, is excellent. The balance of presented and supplementary figures and tables is well-done, with major data presented in the body of the main paper. The English grammar is excellent. I have no major questions for the authors – the paper is currently acceptable for publication, well done.

Minor comments to authors

1.       The NAGL abbreviation should be written in full in the abstract.

2.       Page 3/16 lines 1 and 2 should read “as a predictor of vancomycin-associated…”

3.       Could the authors make one comment about the broader application of cystatin-C estimated renal function beyond paediatrics e.g., should we simple be using cystatin-C for ALL model-informed precision dosing, or is it too early to make this call.

Author Response

Thank you for the thoughtful comments and critiques, which have greatly strengthened the clarity and the overall message of our work. We have responded to this feedback and provide a point-by-point description of the changes we have made below. Specific changes are highlighted with italics below and in the tracked changes version of the manuscript, which accompanies this submission.

Point 1. The NGAL abbreviation should be written in full in the abstract.

  • Thank you, this has been added.

Point 2. Page 3/16 lines 1 and 2 should read “as a predictor ofvancomycin-associated…”

  • This has been fixed.

Point 3. Could the authors make one comment about the broader application of cystatin-C estimated renal function beyond paediatrics e.g., should we simply be using cystatin-C for ALL model-informed precision dosing, or is it too early to make this call.

  • Unfortunately, cystatin c is not routinely available at many pediatric institutions. And, cystatin c results may not return quickly enough to inform TDM decisions if not routinely available. While we have found that cystatin c improves estimation of vancomycin clearance compared to creatinine, most drug development studies have been performed using creatinine-based GFR equations. Thus, additional research may be needed to establish that this is also true of other renally eliminated drugs. We have included the following in the Discussion: “As the availability of cystatin c at pediatric institutions increases, and experience with cystatin c-based eGFR equations mounts, cystatin c may replace creatinine as the biomarker to inform dosing of other renally eliminated drugs as well.”

Reviewer 3 Report

Downes et al. presented a nonparametric-based TDM strategy for optimizing vancomycin therapy. Their work focused on critically ill children and attempted to incorporate markers of acute kidney injury as covariates for clearance. They hypothesized that including them could lead to better AUC predictions. The manuscript is well-written and straightforwardly presents the results, and the conclusions are concise and clear.

There are a few matters the authors could elaborate on:

- have the authors checked if the final model covariates (estimated GFR based on the cystatin c-based Hoek equation & NGAL concentration normalized to urinary creatinine) are correlated? Often, at least in parametric methods, it is ill-advised to include such variables simultaneously.
- The manuscript focuses on establishing the model and the method. Still, it would be interesting to see if the newly built model led to different dose-adjustment recommendations than the log-linear method currently used at the authors' institution. How does the AUC prediction method accuracy compare to the already published methods (or used in the authors' clinical site)? This topic could be the subject of a separate publication, but a short comment would be welcomed.
- Could the authors present how the concentrations changed over time? Such a graph (either as a mean or a point plot) could be included in the supplementary data file.
- Were the optimal sampling times the same for all the subjects, or does the algorithm allow adjusting them according to the known concentrations on a subject-to-subject basis? If the sampling times were optimized, they should be presented as well.
- The authors could include a short statement/opinion on how nonparametric approaches differ from the most widely used parametric ones.
- The authors use the term "imprecision". However, the parameter they use is most widely known as "mean absolute error" (MAE) or "absolute error". (im)Precision is commonly used to describe the spread of the data, and CV is often used as its measure

Author Response

Thank you for thoughtful comments and critiques, which helped us strengthen the clarity and the overall message of our work. We have responded to this feedback and provide a point-by-point description of the changes we have made below. Specific changes are highlighted with italics below and in the tracked changes version of the manuscript, which accompanies this submission.

Point 1. Have the authors checked if the final model covariates (estimated GFR based on the cystatin c-based Hoek equation & NGAL concentration normalized to urinary creatinine) are correlated? Often, at least in parametric methods, it is ill-advised to include such variables simultaneously.

  • This was evaluated and we found that the correlation between the variables was low (-0.27). We have added a statement to clarify this in the Results (section 3.2), “However, the incorporation of the uNGAL on CL, parameterized as the natural logarithm of uNGAL/uCr, led to the largest -2*LL and AIC reductions. The correlation between log uNGAL and eGFRHoek was low (-0.27). No additional covariates were informative on vancomycin CL or V1 and, thus, this model constituted the full model for testing.”

Point 2. The manuscript focuses on establishing the model and the method. Still, it would be interesting to see if the newly built model led to different dose-adjustment recommendations than the log-linear method currently used at the authors' institution. How does the AUC prediction method accuracy compare to the already published methods (or used in the authors' clinical site)? This topic could be the subject of a separate publication, but a short comment would be welcomed.

  • We appreciate this suggestion and, in fact, this is the focus of an ongoing study we are performing at the Children's Hospital of Philadelphia (NCT05691309) in which we are comparing Bayesian AUC to clinical AUC calculations (using log-linear methods). We have clarified that this is an ongoing study in our Discussion: “An ongoing, prospective observational study (NCT05691309) at our institution is evaluating how well Bayesian AUC using our Hoek model aligns with clinical AUC calculations (using log-linear calculations) and assessing the ways in which this would influence dosing recommendations.” We hope that results will be available later this year.

Point 3. Could the authors present how the concentrations changed over time? Such a graph (either as a mean or a point plot) could be included in the supplementary data file.

  • We have added an observed concentrations vs. time after dose graph vs. to the supplemental files (S1) and made reference to its inclusion in section 3.2.

Point 4. Were the optimal sampling times the same for all the subjects, or does the algorithm allow adjusting them according to the known concentrations on a subject-to-subject basis? If the sampling times were optimized, they should be presented as well. 

  • The optimal sampling times were determined separately for each subject and therefore differed. We have added a table displaying the optimal sampling times for the 3 models, using 1 and 2 optimal samples, as a Supplemental Table (S4). This is referenced in section 3.3: “The optimal sampling times relative to the end of the infusion for each of the 3 models are shown in Table S5.”

Point 5. The authors could include a short statement/opinion on how nonparametric approaches differ from the most widely used parametric ones. 

  • Summarizing differences between nonparametric and parametric approaches is not easily done with a short statement. However, we have added a few sentences in the Discussion and included a reference to a thorough review of this topic, which we hope will help readers who are less familiar with both methods.
  • “Another goal of this study was to develop a popPK model that could be used to inform AUC-based vancomycin dosing using Bayesian estimation and limited sampling. We utilized a nonparametric popPK modeling approach, which differs from the more widely used parametric popPK methods in that it assumes that the distribution of parameter values is not necessarily described by an a priori defined continuous function (e.g. normal distribution).43 As a result, more commonly used statistical measures of variability in parametric popPK models (e.g. mean, standard deviation, coefficient of variation) may not fully describe the shape and structure of a non-parametric distribution and, as such, the idea of typical parameter values and interindividual variability around them does not apply. However, because nonparametric approaches allow for parameter probability distributions to take any shape, subpopulations and outliers can be detected, which is ideal in a popPK study of a highly dynamic population such as critically ill children, and methods for Bayesian estimation of individual PK parameters are robust. A more thorough review of nonparameteric and parametric popPK approaches have been published.43   

Point 6. The authors use the term "imprecision". However, the parameter they use is most widely known as "mean absolute error" (MAE) or "absolute error". (im)Precision is commonly used to describe the spread of the data, and CV is often used as its measure.

  • We recognize that the term “precision” is more often used in parametric popPK modeling to describe the spread (ie CV) of the data. However, we believe it is more logically consistent to couple the term "imprecision" with "bias" and "precision" when speaking of "accuracy". We also believe that CV is a less useful measure of imprecision than MAE, SD, mean squared error, or root mean squared error, because unlike all the others, CV necessarily approaches infinity as the true measurement approaches zero. We chose to use MAE as our measure of spread. However, we have added the more common terminology in the methods (section 2.4), and, we have tried to consistently use the terms bias and imprecision throughout the manuscript.
  • “Bias was calculated as (AUCpred – AUCobs )/ AUCobs * 100 and imprecision as |AUCpred – AUCobs| / AUCobs * 100, where AUCpred is the generic AUC24 predicted by our models: AUCfull, AUCHoek and AUCSchwartz. Data were then summarized as the median bias (median percentage predictive error) and imprecision (median absolute percentage predictive error) for each model, as well as the fraction of subjects whose AUCpred was within 5%, 10%, 15%, and 20% of his/her AUCobs.”